# Folate and Vitamin B12 Levels in Chilean Women with PCOS and Their Association with Metabolic Outcomes

**DOI:** 10.3390/nu16121937

**Published:** 2024-06-19

**Authors:** Matías Carrasco-Cabezas, Taís Silveira Assmann, Paz Martínez, Leslie Cerpa, Susan Calfunao, Bárbara Echiburú, Manuel Maliqueo, Nicolás Crisosto, Francisca Salas-Pérez

**Affiliations:** 1Laboratory of Chemical Carcinogenesis and Pharmacogenetics, Faculty of Medicine, Universidad de Chile, Santiago 8320000, Chile; matias.carrasco.c@ug.uchile.cl (M.C.-C.); leslie.cerpa@uchile.cl (L.C.); susan.calfunao@uchile.cl (S.C.); 2Graduate Program in Medical Sciences, Endocrinology, Department of Internal Medicine, Faculty of Medicine, Federal University of Rio Grande do Sul, Porto Alegre 90035-003, Brazil; taisassmann@hotmail.com; 3Laboratory of Endocrinology and Metabolism, Department of Internal Medicine, West Division, Faculty of Medicine, Universidad de Chile, Santiago 8320000, Chile; paz.martinez@uchile.cl (P.M.); bechiburu@uchile.cl (B.E.); mmaliqueo@uchile.cl (M.M.); ncrisostok@gmail.com (N.C.); 4Health Sciences Institute, Universidad de O’Higgins, Rancagua 3070000, Chile; 5Latin American Network for Implementation and Validation of Clinical Pharmacogenomics Guidelines (RELIVAF-CYTED), Santiago 8320000, Chile; 6Endocrinology Unit, Department of Medicine, Clínica Alemana de Santiago, Faculty of Medicine, Universidad del Desarrollo, Santiago 7650568, Chile

**Keywords:** PCOS, MTHFR, metabolism, polymorphism, vitamins

## Abstract

Polycystic ovary syndrome (PCOS) is a common endocrine disorder that affects women of reproductive age. Many women with PCOS have been found to have an unbalanced diet and deficiencies in essential nutrients. This study aimed to assess the levels of folate and vitamin B12 (B12) and their relationship with metabolic factors in women with PCOS. Anthropometric, clinical, and genetic analyses were conducted to evaluate markers related to one-carbon metabolism in women with PCOS and in a control group. The PCOS group had a higher BMI and HOMA-IR (1.7 vs. 3.1; *p* < 0.0001). HDL cholesterol levels were 23% lower and triglyceride levels were 74% higher in women with PCOS. Although there were no significant differences in folate and B12 levels between the PCOS and control groups, over 60% of women with PCOS had low B12 levels (<300 pg/mL) and high homocysteine levels. In addition, the MTHFR A1298C and C677T polymorphisms were not associated with PCOS. Moreover, erythrocyte folate levels were positively correlated with fasting glucose, triglycerides, and free androgen index, and negatively correlated with SHBG and LH levels. These results suggest that B vitamins may be associated with the metabolic phenotype in PCOS. This study emphasizes the potential link between folate, vitamin B12, and metabolic and hormonal outcomes in women with PCOS.

## 1. Introduction

Polycystic ovary syndrome (PCOS) is a prevalent endocrine disorder affecting 10–13% of women of reproductive age worldwide [1]. Its etiology is complex and its clinical manifestations are heterogeneous regarding reproductive, metabolic, and psychological features [2,3]. It has been reported that women with PCOS have several disturbances, including insulin resistance, obesity, and dyslipidemia, which in turn increase the risk of type 2 diabetes, hypertension, cardiovascular disease, and non-alcoholic fatty liver disease, among others [4,5]. The development of PCOS has been linked to numerous genes, including the methylenetetrahydrofolate reductase (MTHFR) gene. This gene encodes an enzyme that plays a critical role in the metabolization of folic acid and is essential for DNA methylation [6,7]. Specifically, MTHFR reduces 5,10-methylenetetrahydrofolate (5,10-methylene-THF) to 5-methyltetrahydrofolate (5-methyl-THF), which produces methyl donors for the conversion of homocysteine (Hcy) to methionine. This enzyme is a critical component of one-carbon metabolism [8]. Several single nucleotide polymorphisms (SNPs) on the MTHFR gene have been described in this context. Among these SNPs, C677T (rs1801133) and A1298C (rs1801131) are the most clinically relevant due to their ability to reduce the MTHFR activity and increase Hcy levels, a risk factor for cardiovascular disease [9]. The C677T single nucleotide polymorphism (SNP) is characterized by a cytosine-to-thymine (C-to-T) substitution, resulting in the conversion of alanine to valine and promoting thermolability, which in turn reduces enzymatic activity. Conversely, the A1298C variant is caused by replacing adenine with cytosine, which changes the glutamate residue to alanine and reduces enzymatic activity [9].

Dietary folate is the natural form of vitamin B9, a water-soluble vitamin, while folic acid (FA) is the synthetic form and is added to staple foods and supplements to fulfill folate requirements. Additionally, the gut microbiota plays a role in folate biosynthesis, which contributes to the overall folate status of the individual [10]. Folates and one-carbon metabolism are pivotal in regulating cell function across all tissues. This encompasses nucleic acid de novo synthesis, amino acid production, and the synthesis of methyl donors, such as S-adenosylmethionine [11]. It has been demonstrated that folates and FA can reduce the risk of neural tube defects. As a result, numerous countries have established mandatory folic acid fortification policies to increase the consumption of folic acid among women of childbearing age [12]. Chile has implemented a mandatory fortification policy since 2000, which is considered a safe, cost-effective, and sustainable intervention. However, despite this success, the theoretical risk persists regarding the interaction between folate and vitamin B12 levels and their potential impact on anemia, cognition, and metabolism. Nevertheless, concerns have been raised regarding the potential harms of inadvertently exposed individuals with low vitamin B12 levels and maternal obesity [13], as well as in susceptible groups [14,15].

Although the evidence is inconclusive, several hypotheses regarding the potential association between elevated folate levels and diminished vitamin B12 status have been proposed. These hypotheses include FA-mediated vitamin B12 oxidation, the methyl folate trap, and malabsorption of supplemented vitamins [16]. In this context, it has been reported that individuals with deficient or inadequate vitamin B12 levels may experience cognitive impairment, possibly due to the inactivation of the active form of B12 (holotranscobalamin) [17,18]. However, the specific concentration of folate that triggers this phenomenon has yet to be identified. These findings underscore the importance of further research into folate levels in the Chilean population.

Due to the central role of folate and vitamin B12 in one-carbon metabolism, an imbalance in their supply may affect DNA methylation and alter the gene expression profile, potentially leading to the development of hormonal disturbances characteristic of the PCOS phenotype. The effects of the C677T and A1298C SNPs on MTHFR activity have been well established in the literature; however, the impact of these SNPs on folate and vitamin B12 levels in PCOS remains uncertain. This study aimed to determine the folate and vitamin B12 status in Chilean women with PCOS, their impact on metabolic parameters, and their association with *MTHFR* SNPs.

## 2. Materials and Methods

### 2.1. Study Design

A total of 58 Chilean women newly diagnosed with PCOS and normoandrogenic controls were recruited at the Unit of Endocrinology and Reproductive Medicine, University of Chile, Santiago, Chile. The current study encompassed women of reproductive age, specifically between the ages of 18 and 34, with a body mass index (BMI) ranging from 18 kg/m^2^ to 35 kg/m^2^. At the initial medical examination, anthropometric measurements were recorded, blood samples were collected, and transvaginal ultrasounds were performed to assess ovarian volume and follicle count. The study was conducted during the early follicular phase (between the third and seventh day of the menstrual cycle). In women presenting with amenorrhea (defined as three or more months without menstruation), blood samples were collected at any time. Clinical hyperandrogenism was determined using the Ferriman–Gallwey score, a clinical assessment tool for hair growth in androgen-dependent sites, including the lip, chin, chest, and others [19]. This case-control study followed the STROBE and STREGA guidelines for reporting genetic association studies [20,21]. The objective was to enhance transparency and reproducibility by providing comprehensive details on the study design, data collection methods, and statistical analysis. This was done to ensure the robustness and reliability of our research findings.

### 2.2. Inclusion and Exclusion Criteria

Inclusion criteria for women with PCOS were as follows: (a) clinical (Ferriman–Gallwey score > 7.0), (b) serum testosterone > 0.8 ng/mL, or androstenedione > 2.46 ng/mL, and/or free androgen index (FAI) > 5.0. In addition to hyperandrogenism, women with PCOS must exhibit oligoovulation or polycystic ovarian morphology on ultrasound. This was evidenced by a follicle count per ovary of ≥20 and/or an ovarian volume of ≥10 mL on either ovary. Furthermore, the absence of corpora lutea, cysts, and dominant follicles was confirmed. Inclusion criteria for control women were the absence of hyperandrogenism and a history of regular menstruation. Exclusion criteria were pregnancy, lactation, vegetarianism, and veganism. Additionally, women with PCOS who had hematologic, infectious, or inflammatory diseases, ischemic coronary disease, diabetes, hypothyroidism, or Cushing’s syndrome were excluded. Furthermore, women who had used antibiotics in the past six months, or hormonal contraception, metformin, methotrexate, or sulfasalazine, as well as medication for epilepsy or hypertension and vitamin supplements, were not included in the study.

### 2.3. Anthropometric and Ultrasound Examinations

All patients underwent laboratory and anthropometric measurements. Body weight (kg), height (cm), and waist circumference (cm) were obtained by a trained technician using validated procedures and BMI was calculated as weight (kg) divided by the square of height (m^2^). Polycystic ovary morphology (PCOM) was determined by transvaginal ultrasound using a transducer (Acuson EV-8C4, Siemens, Erlangen, Germany) by a single experienced technician during the early follicular phase of the menstrual cycle or at any time in women with amenorrhea.

### 2.4. Biochemical Analyses

Following an overnight fast, venous blood samples were collected for subsequent laboratory procedures. Biochemical and lipid profiles, including total cholesterol, high-density lipoprotein cholesterol, and triglycerides, as well as serum folate, red blood cell folate, and vitamin B12, were determined by chemiluminescence (Ortho Clinical Diagnostic, Vitros 4600, Raritan, NJ, USA). Homocysteine was quantified using a fluorometric assay kit (Abcam, Cambridge, MA, USA). Serum folate and vitamin B12 cut-off values for deficiency and insufficiency were defined according to the WHO’s Vitamin and Mineral Nutrition Information System (VMNIS) guidelines [22]. Low-density lipoprotein cholesterol was calculated using the Friedewald formula [23]. Endocrine markers, including total testosterone, androstenedione, estradiol, and 17-hydroxyprogesterone (17-OHP), were analyzed by radioimmunoassay (Gamma counter, Berthold, Bad Wildbad, Germany). Sex hormone-binding globulin (SHBG), follicle-stimulating hormone (FSH), and luteinizing hormone (LH) were quantified by immunoradiometric assay (IRMA) (a Gamma counter, Berthold, Bad Wildbad, Germany was used for SHBG, and a Gamma counter, Packard, Meriden, CT, USA was employed for LH and FSH). Anti-Müllerian hormone (AMH) was analyzed using an enzyme-linked immunosorbent assay (ELISA) (Euroimmun, Lübeck, Germany). The free androgen index (FAI) was calculated using the equation FAI (%) = (total testosterone/SHBG) × 100 [24]. All participants underwent a standardized 2 h oral glucose tolerance test (OGTT-75 g) with insulin measurements after overnight fasting. Blood samples were collected at baseline and 30, 60, 90, and 120 min after the glucose load. Insulin resistance was assessed using the Homeostatic Model Assessment–Insulin Resistance (HOMA-IR) index, which is calculated as follows: fasting insulin (mU/L) × plasma glucose (mmol/L)/22.5. Insulin was measured by chemiluminescence (Centauro XPT, Siemens, Erlangen, Germany). Glucose and insulin plasma concentrations were expressed as an area under the curve (AUC), calculated using the trapezoidal method [25]. The triglycerides and glucose (TyG) index was calculated using the formula Ln (TG [mg/dL] × glucose [mg/dL]/2) [26].

### 2.5. Genotyping Analysis

#### 2.5.1. DNA Isolation

Genomic DNA was extracted from whole blood using an E.Z.N.A^®^ Blood DNA Mini Kit (Omega Bio-tek, Norcross, GA, USA) according to the manufacturer’s protocol with minor modifications to increase the DNA obtained. After extraction, quantification and purity assessment were performed in a Denovix DS-11 spectrophotometer (Denovix Inc., Wilmington, DE, USA). Finally, each DNA sample was diluted to 50 ng/uL for normalization. These samples were stored at −20 °C until further use.

#### 2.5.2. Genotyping

Two SNPs of the *MTHFR* gene, A1298C (rs1801131) and C677T (rs1801133), were analyzed using *TaqMan*^®^ SNP Genotyping Assays (catalog number, 4351379; Thermo Fisher Scientific, Waltham, MA, USA) in an AriaMx real-time PCR system (Agilent Technologies, Santa Clara, CA, USA).

### 2.6. Statistical Analysis

The normal distribution of quantitative variables was tested using Kolmogorov–Smirnov and Shapiro–Wilk tests. Clinical and laboratory characteristics were compared between cases and controls using unpaired Student’s *t*-test, Mann–Whitney test, or chi-square test (χ^2^), as appropriate. Descriptive statistics are presented as mean ± standard deviation (SD) or median (25th–75th percentile values). Categorical variables are expressed as n (%). Variables that demonstrated a significant association with PCOS in the univariate analysis or a biologically relevant association with this condition were selected for inclusion in the multivariate model. Allele frequencies were determined by gene counting, and the Hardy–Weinberg equilibrium (HWE) was tested using the χ^2^ test. Allele and genotype frequencies were compared between groups of subjects using the χ^2^ test. Genotypes were also compared between groups under codominant, recessive, and dominant inheritance models, categorized according to the previous publication [27]. The MORPHEUS web tool generated a correlation heatmap plot (https://software.broadinstitute.org/morpheus; accessed on 23 September 2023). Statistical analyses were performed using the SPSS 29.0.1 software (SPSS, Chicago, IL, USA) and *p* values < 0.05 were considered significant.

## 3. Results

### 3.1. Anthropometric, Clinical, and Biochemical Data of the Sample

A total of 58 women were included in the study, of whom 29 were newly diagnosed with PCOS and 29 were controls. The anthropometric and clinical data of the categorized groups are summarized in Table 1. There was no difference in age distribution between groups, but significant differences were observed in BMI (Controls: 24.1 ± 4.8 vs. PCOS: 32 ± 5 kg/m^2^; *p* < 0.0001), accompanied by increased waist and hip circumferences in the PCOS group. In addition, a slight increase in systolic blood pressure was observed in women with PCOS (Controls: 111.4 ± 10.9 vs. PCOS: 117.1 ± 10 mmHg; *p*: 0.04). Regarding metabolic disturbances, fasting glucose levels were higher in the PCOS group but remained within the biologically normal range (Controls: 82.4 ± 4.8 vs. PCOS: 88.4 ± 11 mg/dL; *p*: 0.006). Furthermore, insulin levels were approximately two-fold higher in women with PCOS compared to controls (Controls: 8.5 ± 5.1 vs. PCOS: 16.6 ± 8.7; *p*: <0.0001), alongside significantly elevated HOMA-IR levels. Moreover, the AUCs for insulin and glucose obtained from the OGTT were also significantly elevated in women with PCOS. Regarding lipid metabolism, HDL cholesterol levels were 23% lower in the PCOS group compared to the control group, whereas triglycerides were 74% higher in women with PCOS. No significant differences were observed between the two groups for total cholesterol, LDL cholesterol, and aspartate aminotransferase (AST) transaminase levels.

### 3.2. Hormonal Parameters of Subjects Included in the Study

The study population was assessed for endocrine markers (Table 2). The Ferriman score was found to be significantly higher in women with PCOS compared to controls (3 (0.5–4) vs. 11 (9–16.5); *p* < 0.0001), as anticipated due to it being an indicator for hyperandrogenism. However, total testosterone levels were similar in both groups. Conversely, sex hormone-binding globulin (SHBG) levels were significantly lower in the PCOS cohort, resulting in a nearly two-fold increase in the free androgen index in this group (*p* < 0.0001). No significant differences were observed between the two groups for androstenedione, FSH, LH, 17-OHP, and AMH.

### 3.3. Folates and Vitamin B12 Circulant Levels

As previously mentioned, an imbalance in folate and vitamin B12 levels could influence one-carbon metabolism. Therefore, serum folate, red blood cell (RBC) folate, and vitamin B12 levels were evaluated in individuals diagnosed with PCOS and in a control group (Table 3). As shown in Table 3, no significant differences were observed between the two groups regarding serum folate, RBC folate, and vitamin B12-complex levels, which remained within the normal range. Nevertheless, cut-off values for vitamin B12 are not well established, but in general terms, >300 pg/mL is considered vitamin B12 adequacy. In our sample, 60.7% of women with PCOS exhibited low-to-normal levels of vitamin B12, thereby increasing their risk of vitamin deficiency. Furthermore, homocysteine levels, a marker of cardiovascular risk, showed an increase of 25% in serum levels in women with PCOS (Controls: 18.6 ± 6.6 vs. PCOS: 23.2 ± 4.3 µmol/L; *p*: 0.01).

### 3.4. Genotype and Allele Frequencies

The presence of specific polymorphisms in the MTHFR gene can result in a reduction in enzymatic activity. Therefore, the allele and genotype frequencies of two single nucleotide polymorphisms (SNPs) on this gene (A1298C and C677T) were investigated to evaluate their frequency between controls and cases with PCOS (Table 4). The frequencies of the two SNPs in both cases and controls were consistent with the expected values for the Hardy–Weinberg equilibrium. Regarding the genotype frequency of the A1298C MTHFR polymorphism, the frequency of the A allele was 0.76 in control subjects and patients. For the MTHFR C677T polymorphism, the allele frequencies in control samples were 0.5 for the C allele and 0.5 for the T allele. In cases, the allele frequencies were 0.52 for allele C and 0.48 for the minor allele T. It is important to highlight that the CC genotype was not identified in the control group. Therefore, applying a statistical test (chi-squared or Fisher’s exact test) was inappropriate for evaluating significant differences between both groups. Furthermore, we sought to assess the potential influence of both polymorphisms on RBC folate levels, considering the three inheritance models: codominant, dominant, and recessive. However, no significant differences were observed between groups, likely due to the limited sample size (Figure 1).

**Table 4 nutrients-16-01937-t004:** Genotype and allele frequencies of SNPs in the *MTHFR* gene.

	Control (*n* = 29)	Cases (*n* = 29)	*p*-Value *
A1298C (rs1801131)			
Allele	0.76	0.76	-
A	0.24	0.24	
C			
Codominant model			
A/A	15 (51.72)	18 (62.07)	0.08
A/C	14 (48.28)	8 (27.59)	
C/C	0 (0)	3 (10.34)	
Recessive model			
A/A + A/C	29	26	0.236
C/C	0 (0)	3 (10.34)	
Dominant model			
A/A	15	18	0.596
A/C + C/C	14	11	
C677T (rs1801133)			
Allele			
C	0.5	0.52	-
T	0.5	0.48	
Codominant model			
C/C	6 (20.69)	8 (27.59)	0.721
C/T	17 (58.62)	14 (48.28)	
T/T	6 (20.69)	7 (24.14)	
Recessive model			
C/C + C/T	23	22	0.999
T/T	6	7	
Dominant model			
C/C	6	8	0.759
C/T + T/T	23	21	

Data are shown as numbers (%) or proportions (*n* = 58). * *p*-values were calculated using chi-squared tests.

**Figure 1 nutrients-16-01937-f001:**
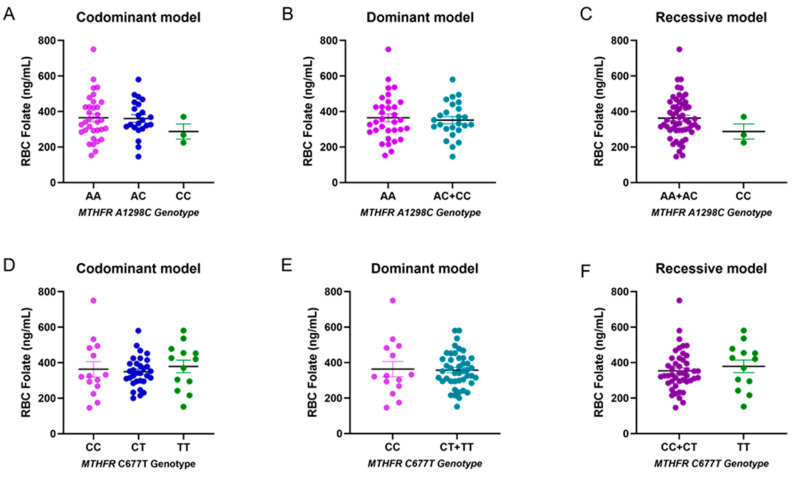
Folate levels according to *MTHFR* genotypes (A1298C and C677T). Panel (**A**–**D**): No statistically significant differences in RBC folate in the codominant model (*n* = 58). Panel (**B**–**E**): No statistically significant differences in RBC folate in the dominant model. Panel (**C**–**F**): No statistically significant differences in RBC folate in the recessive model.

### 3.5. Correlation Analysis

Despite the lack of significant differences between groups in folate and vitamin B12 levels, as shown in Table 3, we investigated potential associations between these vitamins and clinical parameters through correlation analysis in the selected sample of cases and controls (Figure 2). This analysis indicated that red blood cell folate positively correlated with serum folate, an early indicator of low folate status. In addition, RBC folate was positively correlated with fasting glucose, triglycerides, and the TYG index, a marker of insulin resistance. A positive correlation was found between the Ferriman score and the free androgen index regarding hormonal parameters. In contrast, erythrocyte folate was negatively associated with SHBG and LH levels. Serum folate showed no significant correlations. Furthermore, vitamin B12 was positively correlated with LDLc and negatively associated with AMH and 17-OHP levels.

## 4. Discussion

PCOS is a complex endocrine disorder and the mechanisms that contribute to the development of metabolic abnormalities are not fully understood. This study aimed to determine the folate and vitamin B12 status, their impact on metabolic parameters, and their association with *MTHFR* SNPs in Chilean women with PCOS. The results demonstrated that Chilean women with PCOS exhibited hormonal and metabolic disturbances, including elevated free androgen index, BMI, fasting glucose, and dyslipidemia. These findings are consistent with previous studies [28,29,30,31].

Regarding folate and vitamin B12 levels, no significant differences were observed between individuals with PCOS and the control group. In addition, no excessive folate levels were found in serum or red blood cells. Serum folate levels were consistent with previous data from a large sample of women of reproductive age, in which 89.1% of women had adequate folate status. Interestingly, 7% of a sample of the Chilean population exhibited supraphysiological serum folate levels [32]. In addition, the mean serum vitamin B12 concentration in women with PCOS showed insufficient levels alongside elevated homocysteine levels, which could potentially increase methylmalonic acid levels (MMA) [33]. Assessing the potential presence of vitamin B12 deficiency is crucial, as it can lead to significant molecular and cellular alterations. These include the accumulation of oxidative stress, epigenetic modifications, modulation of gene expression or protein content, and changes in lysosomal activity [34]. B12 is a co-factor in converting methyl malonyl-CoA to succinyl-CoA, a substrate in cellular respiration, and is required for hemoglobin synthesis. Moreover, low vitamin B12 has been associated with an accumulation of MMA, an inhibitor of carnitine palmitoyl transferase 1 (CPT1), a key enzyme in the beta-oxidation pathway. Therefore, a reduction in CPT1 activity may explain the increase in lipogenesis and insulin resistance [35,36]. Concerning homocysteine levels, it has been reported that the increased risk of cardiovascular disease is mainly due to their effects on endothelial cells and oxidative stress, mainly by a reduction in PPAR gamma levels [37]. Acquired vitamin B12 deficiency has been reported to result from inadequate dietary intake, malabsorption, and pernicious anemia [38]. In the context of women with PCOS who have a low dietary intake of B12, a corrective intervention to support one-carbon metabolism could potentially improve specific clinical markers, as previously reported [39].

Association analyses revealed a positive correlation between RBC folate, an indicator of long-term folate levels, and fasting glucose levels and triglycerides. These findings are consistent with data from the NAHNES cohort, whereas these associations were not observed with serum folate. Notably, these associations were observed only in younger adults (19–50 y) [40]. Furthermore, a study from India has reported that maternal high folate status and low vitamin B12 may contribute to the development of adiposity and type 2 diabetes [41]. In vitro and animal studies have demonstrated that high FA levels can reduce the activity of several enzymes, including MTHFR, TS, and MTR, and increase homocysteine levels. Additionally, it has been proposed that changes in DNA methylation, upregulation of lipogenesis, and downregulation of glucose transporters, such as GLUT4, may mediate metabolic disturbances due to FA excess [42]. It is essential to highlight that further research is required to elucidate the mechanisms through which obesity exerts differential effects on serum and RBC folate status and their impact on maternal health, particularly in women with PCOS.

Concerning hormonal parameters, a positive correlation was observed between RBC folate, Ferriman score, and free androgen index. Conversely, SHBG and LH demonstrated a negative correlation, suggesting a complex interplay between folate status and hormonal regulation. This is the first study to report these associations between hormonal parameters and B-complex vitamins. A related study examined the effects of plasma vitamin levels in women with PCOS. It was found that niacin exhibited a negative association with SHBG levels [43]. A potential link between folates and androgen excess may be mediated by the folate receptor α (FRα), which has been demonstrated in vitro to be directly activated by testosterone [44].

In the case of vitamin B12, we observed a positive correlation between B12 and LDLc levels, suggesting a potential role for vitamin B12 in lipid metabolism. This is a contradictory finding because low circulating vitamin B12 levels are associated with an adverse lipid profile and inversely associated with LDL-C levels [45]. A prospective cohort study involving adults with diabetes found a significant nonlinear association between serum vitamin B12 levels and cardiovascular mortality. Both low (<369.1 pg/mL) and high (≥506.1 pg/mL) levels were linked to an increased risk of cardiovascular mortality [46]. To validate this finding, it is essential to include a substantial number of samples to ensure a broader range of vitamin B12 levels, especially since our current sample falls within the borderline ranges.

Conversely, we found negative associations between vitamin B12 levels and anti-Müllerian hormone (AMH) and 17-hydroxyprogesterone (17-OHP) levels, indicating a possible relationship between vitamin B12 and reproductive hormones. In studies of healthy premenopausal women, previous research has demonstrated no association between vitamin B12 and AMH and 17-OHP levels [47,48]. However, a study performed in healthy premenopausal women found a slight increase in testosterone levels associated with vitamin B12 intake without changes in other reproductive hormones [48]. This supports the study of this vitamin in PCOS.

To accurately interpret the folate data, the frequencies of two polymorphisms (C677T and A1298C) in the *MTHFR* gene were determined. In our sample, the SNPs C677T and A1298C were found to be in Hardy–Weinberg equilibrium and no significant differences were observed in allele frequency between groups for either SNP. In addition, no association was found between PCOS and the selected polymorphisms and no differences were observed in RBC folate levels according to genotype. The reported prevalence of polymorphisms in the thermolabile variant of *MTHFR* varies considerably among authors. A recent study determined both polymorphisms in a group of young, healthy men and women from northern Chile. In the case of the A1298C and C677T variants, the allele frequencies were similar in comparison to those observed in our population [49]. Additionally, a study conducted in a healthy Chilean mestizo population comprising 146 unrelated subjects, in which the C677T polymorphism was analyzed, showed the following allele frequencies: C = 0.52 and T = 0.48 [50], similar to our control group. The low number of cases exhibiting the minor allele of the A1298C polymorphism, which may account for the absence of differences in blood folate levels, proves the crucial importance of considering the sample size of our study.

In the context of PCOS, several studies have explored the association between PCOS and *MTHFR* SNPs. However, the findings have displayed considerable variability depending on the population studied. Studies in women from Poland, India, Turkey, Korea, and Brazil have found that *MTHFR* polymorphisms, specifically C677T and A1298C, do not appear to influence the risk of developing PCOS [51,52,53,54,55]. In contrast, the A1298C polymorphism has been associated with PCOS in Chinese women, possibly through the modulation of homocysteine levels [56]. Additionally, the C677T polymorphism was associated with PCOS in a separate Chinese cohort, where the T allele increased the risk of PCOS. Furthermore, subjects with the TT genotype showed an increase in insulin resistance in women with PCOS [57]. In particular, meta-analyses suggest that the T variant of C677T is associated with PCOS risk in Asian populations. Additionally, the C allele in A1298C may increase the likelihood of PCOS. Nevertheless, it is essential to note that none of these studies include data from Hispanic populations [9,58]. Furthermore, the TT genotype in the C677T variant has been associated with reduced serum folate levels in the context of controlled supplementation. In addition, the AA allele in the A1298C polymorphism has been linked to increased RBC folate levels [59]. These findings underscore the complex interaction of genetic factors, dietary patterns, and supplementation in influencing folate metabolism in individuals with PCOS.

This study has some limitations. First, the study was conducted at a single center with a relatively small population. Second, the lack of an association between PCOS and the selected polymorphisms may be due to the limited statistical power of the study. Therefore, the results should be validated in a larger, more diverse cohort. In addition, it would be advantageous to determine unmetabolized FA and markers of absorption and metabolization of vitamin B12. These issues constitute limitations in our study. However, this study brings new knowledge regarding the role of B-complex vitamins in PCOS, particularly in metabolic disturbances associated with this syndrome.

## 5. Conclusions

This is the first study to report the folate and vitamin B12 status of women with PCOS, along with their correlations with hormonal parameters, B-complex vitamins, and the allele frequencies of *MTHFR* polymorphisms. These findings suggest a potential link between RBC folate levels and metabolic factors, highlighting the importance of folate metabolism in metabolic health. Further research with a large cohort of participants is needed to validate the influence of these genetic variants. This expanded investigation will provide comprehensive insights into how these polymorphisms contribute to PCOS’s metabolic and hormonal disturbances.

## Figures and Tables

**Figure 2 nutrients-16-01937-f002:**
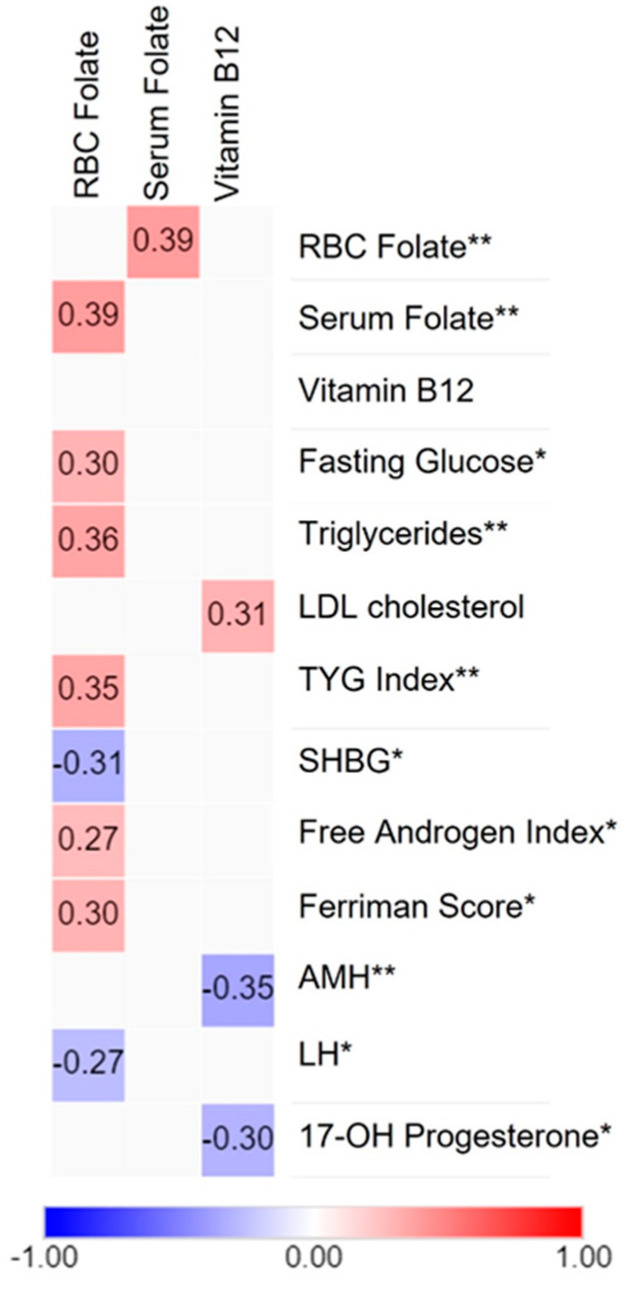
Correlation between clinical parameters and one-carbon metabolism markers in Chilean women. Heatmap showing the association between folates and vitamin B12 (in columns) and metabolic and hormonal parameters (in rows). Positive correlations are shown in red and negative correlations in blue, whereas the absence of correlations is represented in light gray (*n* = 58). The degree of significance is marked as follows: (*) *p*-value < 0.05 and (**) *p*-value < 0.01. This figure was created using the MORPHEUS software, version 2.3.

**Table 1 nutrients-16-01937-t001:** Anthropometric and clinical data of the study population.

Parameter	Control (*n* = 29)	PCOS (*n* = 29)	*p*-Value *
Age (years)	25.8 ± 4.8	25.5 ± 4.2	0.797
Ferriman score	3 (0.5–4.0)	11 (9–16.5)	<0.0001
BMI (kg/m^2^)	24.1 ± 4.8	32 ± 5	<0.0001
WC (cm)	75.7 ± 9.8	92.9 ± 13.4	<0.0001
HC (cm)	87.8 ± 9.4	104.1 ± 14.9	<0.0001
W-H ratio	0.86 ± 0.1	0.87 ± 0.1	0.322
SBP (mmHg)	111.4 ± 10.9	117.1 ± 9.9	0.044
DBP (mmHg)	69.9 ± 6.5	70.6 ± 8.37	0.706
Fasting Glucose (mg/dL)	82.4 ± 6.8	88.4 ± 10.8	0.006
Glucose AUC	12,495 (9825–13,478)	14,430 (12,480–16,733)	0.0002
Insulin AUC	5564 (4038–8338)	11,390 (7771–14,985)	<0.0001
Fasting Insulin (mU/L)	8.5 ± 5.1	16.6 ± 9.7	<0.0001
HOMA-IR index	1.7 (0.85–2.35)	3.1 (2–4.9)	<0.0001
Total Cholesterol (mg/dL)	161.5 ± 30.5	172.1 ± 32.5	0.207
HDL-Cholesterol (mg/dL)	56.2 (40.4–65)	43.4 (36.3–52.1)	0.05
LDL-Cholesterol (mg/dL)	89.5 (65.5–89.5)	102 (75.6–114.5)	0.425
Triglycerides (mg/dL)	84.7 ± 38.4	147.5 ± 165.4	0.005
LDH (U/L)	165.9 ± 37	177.4 ± 32.8	0.107
AST (U/L)	27.7 ± 9.9	31.6 ± 12.9	0.247

Variables are shown as mean ± SD or %. Variables with skewed distribution are shown as median (25th–75th percentile values). *p*-values were calculated using Student’s *t*-test or Mann–Whitney test as appropriate; * *p* < 0.05 was considered statistically significant. BMI: body mass index; WC: waist circumference; HC: hip circumference; W-H: Waist-hip ratio; SBP: systolic blood pressure; DBP: diastolic blood pressure; AUC: area under curve; HOMA-IR index: Homeostatic Model Assessment–Insulin Resistance index; LDH: lactate dehydrogenase; AST: aspartate aminotransferase.

**Table 2 nutrients-16-01937-t002:** Hormonal parameters of the study population.

Parameter	Control (*n* = 29)	PCOS (*n* = 29)	*p*-Value *
Testosterone (ng/mL)	0.56 (0.44–0.7)	0.65 (0.5–0.8)	0.13
SHBG (nmol/L)	43.6 (36.3–70.5)	27.3 (19–35.9)	<0.0001
FAI (%)	4.34 (2.5–5.1)	8.1 (6–12.7)	<0.0001
Androstenedione (ng/mL)	3 (2.3–4.2)	3.6 (2.7–4.9)	0.13
FSH (mUI/mL)	5.8 (5.1–7.6)	5.4 (4.2–6.1)	0.06
LH (mUI/mL)	4.3 (3.2–6.1)	4.3 (2.8–6.1)	0.99
Estradiol (pg/mL)	64.7 (51.6–84.6)	68 (51.3–94.6)	0.63
17-OHP (ng/mL)	1.3 (0.98–1.55)	1.3 (1–1.6)	0.95
AMH (ng/mL)	3.9 ± 2.6	5.0 ± 3.3	0.16

Variables are shown as mean ± SD or %, as appropriate. Variables with skewed distribution are shown as median (25th–75th percentile values). * *p*-values were calculated using Student’s *t*-test or Mann–Whitney test as appropriate; *p* < 0.05 was considered statistically significant. SHBG: sex hormone-binding globulin; FAI: free androgen index; FSH: follicle-stimulating hormone; LH: luteinizing hormone; 17-OHP: 17 hydroxyprogesterone; AMH: anti-Müllerian hormone.

**Table 3 nutrients-16-01937-t003:** One-carbon metabolism parameters in the study population.

Parameter	Control (*n* = 29)	PCOS (*n* = 29)	*p*-Value *
Serum Folate (ng/mL)	12.1 ± 3.2	13 ± 4.9	0.353
RBC Folate (ng/mL)	334.6 ± 114.3	383.7 ± 114.1	0.107
Vitamin B12 (pg/mL)	301 (247–355)	275 (235–347)	0.455
Homocysteine (µmol/L)	18.6 ± 6.6	23.2 ± 4.3	0.01

Variables are shown as mean ± SD. Variables with skewed distribution are shown as median (25th–75th percentile values). * *p*-values were calculated using Student’s *t*-test or Mann–Whitney test as appropriate; *p* < 0.05 was considered statistically significant. RBC folate: red blood cell folate.

## Data Availability

The data that support the findings of this study are available within this article. Further information analyzed during the current study are available upon reasonable request.

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
