# Peer review of "Folate and Vitamin B12 Levels in Chilean Women with PCOS and Their Association with Metabolic Outcomes"

_nutrients, 2024, doi:10.3390/nu16121937_

Round 1

Reviewer 1 Report

Comments and Suggestions for Authors

Manuscript very interesting. The problem of polycystic ovary syndrome (PCOS) has long been widely reported. It is associated with a number of both endocrine and micronutrient or vitamin disorders. Deficiencies of vitamin B12 or folate affect all groups of patients, and are particularly age-related in menopausal women, geriatric patients and, of course, in PCOS.  In the manuscript, the study was conducted with only 29 subjects.  One could, of course, see differences between the study groups, but nevertheless this is a small number of subjects. It is worth enlarging the number of subjects in each group. Correlations between hormones and other parameters very interesting give and promising results. Vitamin B12 should be included in all patients, after prior evaluation of the level of this vitamin. There are also serious disorders of other B vitamins in PCOS. Therefore, it would be advisable to include a B-vitamin complex in addition to hormonal treatment and Vitamin D3.  Tables presented legibly. References I suggest to expand and literature from recent years 2022, 2023. 

Author Response

Please see the attachment. Thanks for all the comments. 

Reviewer 2 Report

Comments and Suggestions for Authors

In the manuscript “Folate and vitamin B12 levels in Chilean women with PCOS and their association with metabolic outcomes”, the authors proposed to evaluate the folate and vitamin B12 (B12) status and its correlation with metabolic parameters in women with PCOS. The subject of the paper is interesting and worths discussion. Nevertheless there are some issues that must be revised. The authors are encouraged to introduce a more mechanistic-orientated discussion. Many subjects should be discussed in a more straightforward way avoiding vague terms. All sections need careful revision.

Specific comments:

1. Abstract needs extensive revision. For instance, the groups are not well explained. In addition, the data is presented with generic terms without a clear presentation of the level of differences detected. A 10% change is very different from a 70%. Please provide a clear abstract.

2. Results description should be numeric as it highlights the differences. In addition, the rationale for the experimental approach should be a bit more present in this section.

3. Study limitations should be included. A take home message should also be present.

Author Response

Please see the attachment. Thanks for the comments; they were very useful and improved the manuscript. 
